# How common depictions of wealth distributions can bias people to underestimate inequality

Jonathan E. Bogard [1] ✉, Colin West[2] & Craig R. Fox [2]

Economic inequality is a critical issue in elections and legislating, so having an informed electorate is vital to sensible policymaking. Unfortunately, misperceptions about the true extent of inequality abound. Here, we demonstrate that the most prevalent formats for representing economic distributions tend to dampen viewer impressions of inequality. We begin by showing that the most popular newspaper portrayals of inequality contain potentially biasing features. Next, in nine experiments (N = 3599 U.S. adults), we explore how these common representations of inequality evoke biased impressions and we test methods to mitigate these biases. Specifically, we document two biases in how people evaluate economic distributions. First, people are under-sensitive to differences in the size of identified population groups (e.g., top 1% vs. top 10%) such that their judgments become distorted by arbitrary ways in which a population is divided (*partition dependence*). Second, people's judgments under-weight information about intermediate groups relative to the most- and least-wealthy groups (*middle neglect*). We test ways to mitigate these biases and conclude by proposing presentation guidelines that promote more accurate impressions.

Economic inequality within industrialized countries has increased considerably over the past half-century and its effects have been linked to negative outcomes including poorer public health[1,2], higher rates of violence[3], increased support for authoritarian leadership[4], and diminished resilience to global pandemics[5]. As a result, economic inequality has become a central issue in many elections and is frequently discussed in the popular media. People's support for policies that ameliorate inequality is shaped by their notions of fairness[6], their beliefs about the causes of inequality[7–9], and their political affiliation. Critically, their support for such policies is also shaped by their subjective impression about the current degree of inequality[10,11]. In this article, we investigate how the methods chosen to represent economic distributions can influence people's subjective impressions of (un)fairness and their support for policies that redistribute wealth.

Despite all the attention that economic inequality has garnered, most people drastically underestimate the extent of inequality in their own country and around the world[11]. Moreover, people's impressions of inequality are influenced by their subjective, provincial impressions of inequality cues, which may not accord with objective reality[12–17]. For instance, when we asked Americans about the current state of economic inequality in United States, the median estimate implied that an average member of the wealthiest 10% of the population holds about 13.5 times as much wealth as an average member of the remaining 90% of the population (see Study 3 below). The true ratio is about 27:1, meaning inequality is 100% larger than respondents estimated on average. Similar errors have been documented among people from around the world[11]. Such pervasive misperceptions can shape the outcome of elections and reduce popular support for policies aimed at addressing inequality[18–20]. In this article, we demonstrate that distorted perceptions of inequality may, in part, result from the ways in which information about economic distributions is commonly described by popular media outlets, politicians, and social scientists.

When describing how any finite resource is distributed across a large group of people—such as the distribution of profits within an

[1]Washington University in St. Louis, St.Louis, MO, USA. [2]UCLA, Anderson School of Management, Los Angeles, CA, USA. ✉e-mail: bogard@wustl.edu

organization or wealth between countries—one must make several choices about how to convey this information. For instance, consider a journalist interested in reporting information about national economic inequality in a popular news article. First, because it would be impractical to list the amount held by every individual in the country, the journalist must summarize this information by dividing the population into groups. This could be done evenly (e.g., four equal quartiles) or unevenly so as to highlight the standing of particular groups (e.g., the wealthiest 1% or the least wealthy 10%). Second, the journalist would have to choose a metric for reporting this information, such as the percentage of wealth held by each group or the amount of wealth held by an average member of each group. Last, the journalist must decide whether to present this information in prose, tables, or graphics. Indeed, there are countless ways in which these factors could be combined to describe a resource distribution, and in the present investigation we show that seemingly inconsequential design decisions can have an outsized impact on how people evaluate this information. Unfortunately, as we will demonstrate, nearly all media outlets choose methods for characterizing wealth distributions that lead readers to form biased impressions that underestimate the true extent of inequality.

To assess the prevalence of potentially biasing methods, we conducted an audit of every article mentioning "economic inequality" in the year 2020 among the eight most popular newspapers in America (see Supplementary Information for more details). Across 2,958 articles, we cataloged the most common ways of presenting information on economic distributions, identifying three patterns that might lead readers to form systematically biased impressions about inequality. First, most of these articles (74%) compared unequally sized population groups, explicitly drawing attention to the extremes of the distribution (e.g., comparing the wealthiest 1% versus the remaining 99%). Second, 62% of articles used an aggregate metric (e.g., percentage of total wealth held by each group) rather than a scaled metric (e.g., amount of wealth held by an average member of each group). And third, 98% of articles contained only textual or tabular representations of the data rather than visual graphs or figures. As we will argue, each of these decisions can lead to systematic underestimation of inequality. An example of all three of these potentially biasing features comes from the Washington Post, which reported that, "The top 1 percent own about a third of the nation's wealth, near the 30-year high for that population. The poorer half of the country, meanwhile, claim roughly 2 percent of the overall wealth."[21] Moreover, these patterns are not limited to the popular press—information on economic distributions is presented using similar formats by highly cited academic articles (e.g., Norton and Ariely[22]; Piketty et al.[23]) as well as reports produced by major institutions (for instance, by the World Inequality Report, Chancel et al.[24]; the US Census Bureau, Fontenot et al.[25]; and the Pew Research Center, Menasce Horowitz et al.[26]). In this article we show that these methods of depicting economic distributions are misleading and generally cause readers to underestimate inequality. Disseminating such systematically misleading information, even unintentionally, can have far-reaching consequences for the well-being of democratic societies.

To illustrate why these formatting choices may be problematic, consider the following description of economic inequality that might be found in the popular media: The wealthiest 10% of Americans currently hold approximately three times the amount of wealth held by the rest of the country. Now ask yourself how fair or unfair this seems to you, and how strongly you would favor redistributing wealth from wealthier to less wealthy people. This description involves all of the biasing features found to be common in our audit of media articles discussing economic inequality: It compares unequally sized groups using an aggregate metric without a visual aid. Now consider an alternative depiction of the same information: The average individual among the wealthiest 10% of Americans holds 27 times as much wealth as an average individual among the rest of the country. These two descriptions are informationally equivalent, but the second description—which we will argue uses a more appropriate metric—gives readers an impression of greater inequality. To wit, in Study 1 (below), we find that representing the same information in this latter format significantly increases support for economic redistribution (compare Fig. 1B, "Average Wealth" to Fig. 1A, "Total Wealth").

The observation that people exhibit inconsistencies when evaluating economic distributions should not be surprising given the cognitive complexity involved. People must compare multiple population groups without any obvious benchmarks or reference points for what is fair. Meanwhile, people's intuitions about resource distributions are presumably adapted for living in small groups rather than large-scale economies involving trillions of dollars spread across millions of households[27,28]. Past research has found that when confronted with complex information, people often reduce the cognitive effort required by considering only a subset of the given information and by simplifying how this information is weighted[29]. Accordingly, we propose that when evaluating economic distributions, people reduce complexity in the following three ways. First, they reduce the information considered by focusing primarily on the amount of wealth held

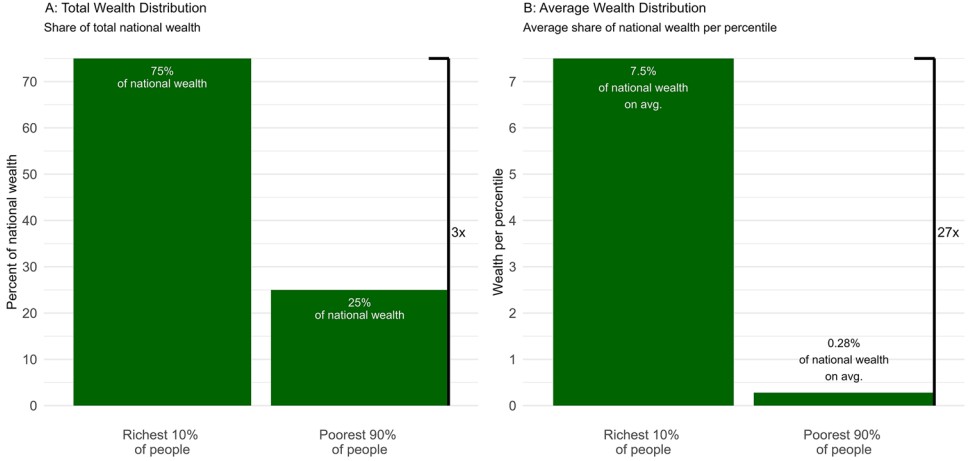

**Fig. 1 | Study 1 Stimuli: Effects of Partition-invariant Metrics.** In Study 1, we presented participants with informationally equivalent representations of the current approximate wealth distribution in the United States, described in terms of either total wealth (**A**) or average wealth per percentile (**B**). Participants rated inequality as less pronounced—and less objectionable – when presented as in (**A**) than compared to (**B**). Thus, rescaling the data in (**B**) to account for unequal bin sizes increases recognition of inequality.

by identified groups while paying less attention to differences in group size. Second, they simplify the weighting of information by comparing a given distribution against a benchmark of equal allocation across identified groups – consistent with the application of an "equality heuristic."[30] Third, they reduce attentional demands by disproportionately focusing on comparisons between the most salient groups, which are generally at the extremes of the distribution (i.e., the top and bottom fractiles).

This account implies two testable hypotheses. First, when people make judgments about the fairness of economic distributions they will be under-sensitive to differences in population group sizes and, consequently, their judgements will vary systematically depending on how the population happens to be divided ("partition dependence"). Second, judgments of fairness will be relatively insensitive to variations in the amount of wealth held by intermediate groups compared to the extreme groups ("middle neglect"). In this paper we provide evidence of both partition dependence and middle neglect, and we test strategies for displaying information about economic distributions that can help overcome these biases.

We present nine experiments ($N = 3599$), demonstrating that the most common methods of presenting economic distributions can distort people's perceptions about the extent of inequality as well as what constitutes an ideal wealth distribution. We assert that these distortions may arise from the use of simplifying strategies when people process information about resource distributions. Participants provided informed consent in all studies, and the research protocol was approved by the UCLA Institutional Review Board. All materials, data, code, and pre-registrations are available in a ResearchBox collection, available online[31]. All studies except for Study 8, a descriptive survey, were pre-registered, and there were no deviations from pre-registration plans. Unless otherwise noted, the statistical tests reported herein represent the results of an Ordinary Least Squares regression test with pre-registered controls. When applicable, tests are all two-sided.

## Results

### Partition dependence in evaluations of economic distributions

We propose that people assess the fairness of resource distributions against a natural benchmark of equality between explicitly identified groups (i.e., people ask themselves, "How much does this economic distribution deviate from perfect equality across groups that are presented?"). If people rely, to any extent, on this kind of equality heuristic when evaluating the fairness of a given economic distribution and if they are also under-sensitive to difference in group sizes, then their judgements will vary systematically with the arbitrary partitioning of the population – that is, the number and relative sizes of the groups into which the population happens to be divided. Thus, reliance on an equality heuristic does not by itself give rise to partition-dependent preferences but, when combined with under-sensitivity to differences in group sizes, preferences will vary systematically with the way the population happens to be subdivided. (For a related discussion of how methods for measuring beliefs about inequality may influence judgments about economic mobility, see Swan and colleagues as well as the response from Davidai and Gilovich[32,33]).

Study 1 ($N = 391$; $n_{control} = 195$, $n_{treatment} = 196$) provides initial evidence supporting the hypothesis that people's judgments about inequality are partition dependent. When the wealth distribution of an unidentified country was represented using an aggregated metric of total wealth to each fractile (Fig. 1A), we found that just 56% of respondents supported a policy to redistribute wealth. In contrast, when the same distribution was presented using the scaled metric of average wealth per percentile (as in Fig. 1B), more than 70% of participants supported the same policy. While the two distributions were informationally identical, participants' perceptions of the need for redistribution was consistent with a bias toward between-group

| What percent of the United States' total wealth should the following groups control? | |
|---|---|
| **A** | **B** |
| Richest 20% | Richest 1% |
| Next 20% | Next 4% |
| Next 20% | Next 15% |
| Next 20% | Next 30% |
| Poorest 20% | Poorest 50% |

**Fig. 2 | Study 2 Stimuli: Evenly *versus* Unevenly Divided Population Groups.** In Study 2, participants constructed ideal wealth distributions using either five evenly divided groups (**A**) or five quasi-logarithmically divided groups (**B**). We found that participants are under-sensitive to the width of population bins when constructing these distributions, implying a preference for greater inequality in the quasi-logarithmic than the even-partition condition.

equality and under-sensitive to differences in group sizes ($\chi^2(1) = 8.23$, $p = 0.004$, $b = 0.16$, 95% CI = [0.05, 0.26]). The scaled metric (average wealth per percentile) automatically accounts for such an under-sensitivity and thus, arguably, represents a more neutral metric for assessing people's perceptions of inequality. See Supplementary Information for detailed results.

In Study 2, we demonstrate partition dependence by asking 165 Americans to characterize how they think wealth ideally should be distributed in the United States. We randomly assigned participants to one of two versions of this task, partitioning the population into five groups either as equally sized quintiles ($n = 82$) or quasi-logarithmically ($n = 83$; Fig. 2A, B). In the quasi-logarithmic condition, "unpacking" the top 20% of the population into three sub-groups (Fig. 2B) resulted in an increase in the share of total national wealth that participants said the top quintile ought to hold from 27% to 48% ($t(159) = 6.96$, $p < 0.001$, $b = 21.5$, 95% CI = [15.4, 27.6]).

We found a similar pattern of results when we asked participants to estimate the current wealth distribution in the United States ($t(159) = 7.42$, $p < 0.001$, $b = 28.5$, 95% CI = [21.0, 36.0]), suggesting that the impact of partitioning extends from normative preferences to descriptive estimates (Eriksson & Simpson, 2012[34]; *cf.* Norton and Ariely[22]). See Supplementary Information Table S2 for detailed results.

These results are consistent with partition dependence such that people's judgments are biased towards a benchmark of equal allocation across identified groups with insufficient sensitivity to the relative sizes of these groups. As a result, participants appear to be more tolerant of inequality when the top of the distribution is divided into a greater number of subgroups, as in the quasi-logarithmic condition.

We note that such quasi-logarithmic population partitioning is a common method of representing economic inequality in the mainstream media, presumably because many journalists wish to highlight the large concentration of wealth at the very top of the distribution. However, by partitioning the wealthiest groups into a greater number of subgroups (or smaller-sized groups), they ironically lead readers to perceive the wealth distribution as *less* concentrated.

In Study 3, we tested the robustness of partition dependence. We asked 162 Americans ($N_{50-50} = 73$, $N_{unpacked} = 89$) to report their estimated and ideal wealth distributions for the United States using alternative partitioning schemes, unpacking the top of the wealth distribution into evenly spaced population groups—either {50-50} or {10-10-10-10-10-50}. As predicted, participants allocated significantly more wealth to the top half of their ideal distribution when it was subdivided into five groups ($M = 68\%$) compared to when it was a single group ($M = 60\%$; $t(160) = -3.16$, $p = 0.001$, $b = 7.64$, 95% CI = [3.10, 12.18]). Again, we also found a similar effect for *descriptive* views of how participants thought wealth was currently distributed (see Supplementary Information Table S3 for details).

In Study 4 ($N = 475$; $N_{deciles} = 235$, $N_{quartiles} = 240$), we found that partition dependence for ideal distributions persisted even when we provided participants with the actual current U.S. wealth distribution in a 50-50 partitioning as a point of reference. Participants allocated

significantly more wealth on average to the top half of the distribution when confronted with a {10-10-10-10-10-50} partition ($M = 73\%$) compared to a {25-25-50} partition (66%; $t(473) = 4.40$, $p < 0.001$, $b = 6.40$, 95% CI = [3.55, 9.25]). As before, a seemingly inconsequential decision about how to group the population when eliciting preferences led to an increase in participants' apparent tolerance of inequality.

One obvious way to mitigate partition dependence is to present information on economic distributions using equally sized population groups (e.g., five quintiles or ten deciles). Indeed, this makes direct comparisons between groups straightforward and circumvents the problem of partition dependence. However, using equally sized population groups has drawbacks as it masks the differences that might exist within groups, particularly at the top of the distribution. For instance, highlighting the fact that the wealthiest 1% of the US population holds 35% of total national wealth hides the fact that most of this wealth is concentrated within the wealthiest decile of that population sub-group: The wealthiest 0.1% holds 20% of national wealth[35]. Thus, there are good reasons to compare population groups of different sizes, but in order to avoid systematic misperceptions, these must be accompanied by metrics that adjust for differences in group size. This can be accomplished by presenting economic distributions using partition-invariant metrics such as "average wealth per individual" in a given group or "percent of national income per percentile" for each group. In the next study, we tested whether such scaled metrics could reduce or eliminate partition dependence when people evaluate how wealth is distributed across population groups of different sizes.

In Study 5, we randomly assigned 402 participants to one of four conditions in a 2 (population partition: quintiles vs. quasi-logarithmic) × 2 (metric: average wealth per percentile vs. percentage of total wealth) experimental design ($N_{quint-pct} = 99$, $N_{quint-ttl} = 109$, $N_{log-pct} = 93$, $N_{log-ttl} = 101$). We presented each participant with the wealth distribution of an unidentified country (approximating the current distribution of the United States) using one of these four presentation formats. Participants judged the fairness of the wealth distribution then rated their preferences for redistribution in this society. As in the previous studies, when presented with the metric of "percentage of total wealth," participants' fairness judgements significantly varied with the partitioning of the population groups ($t(398) = -3.30$, $p = 0.001$, $b = -0.95$, 95% CI = [-1.52, -0.38]). On the other hand, there was no statistically significant simple effect of partitioning when using the metric of "average wealth per percentile" ($t(398) = 1.77$, $p = 0.077$ NS, $b = 0.53$, 95% CI = [-0.06, 1.12]; See Supplementary Information Table S4 for the full set of results.) Thus, we observe a significant interaction such that the effect of quasi-logarithmic partitioning on fairness judgements is significantly attenuated when using an average wealth metric compared a total wealth metric ($t(398) = 3.56$, $p < 0.001$, $b_{interaction} = 1.49$, 95% CI = [0.66, 2.32]). This suggests that economic inequality ought to be represented using partition-invariant metrics, especially when making comparisons between unequally sized population groups.

Taken together, the five studies presented thus far suggest that people may reduce the complexity inherent in evaluating economic distributions by starting from a consideration of equal allocation between groups, with insufficient adjustment depending on group size. This cognitive strategy can lead to pervasive biases when applied to evaluations of large-scale economic distributions over unequal groups using metrics that vary with group size (such as total wealth), as often presented by journalists, researchers, and policy makers.

## Middle neglect in evaluations of economic distributions

We have proposed that people reduce the complexity inherent in evaluating economic distributions by not only benchmarking against even allocation but also by disproportionately attending to the most salient groups. Societal groups that are at the top and bottom of the wealth and income distributions (i.e., groups that are the most and least wealthy) generally command the greatest attention due to their symbolic importance[36] and because they are usually listed first and last (garnering further salience due to primacy and recency). As a result, people may be systematically under-sensitive to the amount of a given resource held by intermediate groups. Thus, we argue that a bias to give greater weight to extreme fractiles in the distribution results not from a principled disregard of the middle class (*cf.* Davidai & Deri, 2019[37]) but rather from a tendency to direct attention toward more salient information—in this case, the most and least wealthy groups. We pause to emphasize that "middle neglect" refers to the claim that people are under-sensitive to information about intermediate fractiles (relative to the first and last fractiles), not that they are entirely insensitive to them. Thus, for instance, anything that heightens the salience of, or draws explicit attention to, these middle fractiles will likely attenuate (or may even reverse) middle neglect. Indeed, it is common among media and policy reports of inequality to ignore intermediate groups entirely, for example: "The top ten percent of households own 76% of all wealth in the U.S., while the bottom 50% of households own just 1% of all wealth."[38] Note that, in this example, attention is drawn to both the wealthiest group as well as the bottom half of the wealth distribution, while attention is drawn away from the welfare of everyone in between. In the next four studies, we investigate the phenomenon of middle neglect in evaluations of economic distributions, and we measure the effects on people's judgements of fairness and support for policies that redistribute wealth.

In Study 6, we tested for middle neglect by asking 197 participants to rate the fairness of eight hypothetical economic distributions (within-participant, repeated measures). Across these eight distributions, we manipulated the average income of three population segments. The wealthiest quintile of society was described as having an average annual income of either $220,000 or $440,000; the middle quintile was described as having an average income of either $60,000 or $120,000; and the bottom quintile was described as having an average income of either $14,000 or $28,000. Participants observed and rated all eight of the possible combinations of incomes (three population groups with two possible average income levels each equals $2^3$ possible combinations).

The Ordinary Least Squares regression results support our prediction of middle neglect. Doubling the average income of the wealthiest group led respondents to rate a distribution as significantly less fair ($t(193) = -7.26$, $p < 0.001$, $b = -0.730$, 95% CI = [-0.93, -0.53]), and doubling average income of the least-wealthy group led respondents to rate a distribution as significantly more fair ($t(193) = 9.61$, $p < 0.001$, $b = 1.199$, 95% CI = [0.95, 1.45]). However, doubling the average income of the middle group of society had no significant impact on fairness judgments ($t(193) = 1.68$, $p = .094$, NS, $b = 0.128$, 95% CI = [-0.01, 0.14]), despite the objective increase in equality as measured by a Gini coefficient. A *post hoc* linear hypothesis test of the coefficients revealed that the effect of doubling the income of the middle was significantly less consequential for fairness evaluations than doubling the average income of either the wealthiest ($F = 42.584$, $p < 0.001$) or the least wealthy ($F = 56.319$, $p < 0.001$) groups. This suggests that people are relatively insensitive to welfare changes among middle groups compared to extreme groups.

A question remains, however, whether middle neglect is ideological or unintentional. That is, do people disregard information about middle groups because they have a principled view that the extreme groups are more important? Or is middle neglect instead caused by a phenomenon akin to a "cognitive blind spot," leading people to underweight information about the middle regardless of their ideological concerns? To better understand the mechanism, we measured participants' stated priorities when evaluating economic distributions. Participants rated the following four statements using a −7 (completely disagree) to +7 (completely agree) scale: (1) "We ought to make sure

that the POOREST members of society are doing as well as they can" ($M = 4.4$, SD = 3.2); (2) "We ought to reduce the amount of inequality between the TOP and BOTTOM groups in society" ($M = 3.19$, SD = 3.9); (3) "We ought to make sure that it is possible to become EXTREMELY WEALTHY to incentivize people" ($M = 0.4$, SD = 4.5); (4) "We ought to build as strong, robust and wealthy of a MIDDLE class as possible" ($M = 4.1$, SD = 3.0). When we control for participants' responses to these measures, we still observe middle neglect in their fairness judgments of the eight economic distributions (see Supplementary Information Tables S6–S9 for further details). Furthermore, consistent with the notion of a "cognitive blind spot" for the middle of the income distribution, we find that participants' stated concern for the middle class has no significant effect on their sensitivity to doubling the average income of the middle group ($t(187) = 1.12$, $p = 0.264$ NS, $b_{interaction} = 0.013$, 95% CI = [−0.011, 0.037]). In contrast, as expected, we find that stated concern for people who are poorest moderates the effect of doubling the poorest group's average income on fairness judgements ($t(187) = 5.01$, $p < 0.001$, $b_{interaction} = .080$, 95% CI = [0.05, 0.11]), and stated concern for the wealthy moderates the effect of doubling the wealthiest group's average income ($t(187) = 3.59$, $p < 0.001$, $b_{interaction} = 0.038$, 95% CI = [0.02, 0.06]). These findings suggest that middle neglect reflects a bias in information processing rather than simply a principled belief that the wealthiest and least wealthy groups should carry more weight in evaluations of economic distributions.

In Study 7, we investigate this mechanism more directly by using visual cues to redirect participants' attention when they evaluate economic distributions. If people are in fact biased toward focusing on the extremes of distributions (i.e., the groups that are most and least wealthy)—for whatever reason—then increasing the visual salience of these groups should have little additional effect on people's fairness judgments. Conversely, visual cues that highlight the middle of an economic distribution should have a relatively larger effect on fairness judgments, but only if people tend to unintentionally neglect this information (i.e., if it is a "cognitive blind spot"). However, if middle neglect is instead driven by an ideological preference for disregarding the intermediate groups, heightening their visual salience should have no differential impact compared to highlighting the extremes.

We asked 868 participants to rate the fairness of sixteen hypothetical economic distributions across four quartiles displayed in a tabular format. We randomly assigned participants to one of three between-participants conditions in which we visually highlighted in yellow (1) both the wealthiest and least wealthy groups (i.e., the top and bottom rows in the table; $N = 294$), (2) both of the middle groups (i.e., the upper-middle and lower-middle rows; $N = 283$), or (3) none of the groups ($N = 291$). As predicted, highlighting the extreme groups had a only small effect on fairness judgements, whereas highlighting the middle groups had a much larger effect ($b = 0.08$, Bootstrapped 95% CI = [0.05, 0.11]; see Supplementary Information section 8 for details on this analysis). These results are consistent with a "cognitive blind spot"—rather than an ideological—explanation for the phenomenon of middle neglect.

In Study 8, we further examined middle neglect by simply asking people to report how they evaluated information about a given economic distribution using an empirical approach adapted from research on Query Theory[39]. We asked 269 participants (within-participants) to rate the fairness of a wealth distribution that (unknown to them) approximated the current level of inequality in the United States, and to indicate their level of support for a policy to redistribute wealth. Participants then listed the thoughts that occurred to them as they made their evaluation, and subsequently (after listing all thoughts) categorized each of their own thoughts using the labels we provided them: "I focused on how much of the total wealth was concentrated at the TOP of the distribution"; "I focused on how much of the total wealth was at the BOTTOM of the distribution"; "I focused on the

DIFFERENCE between the amount wealth at TOP versus the BOTTOM of the distribution"; "I focused on how much of the total wealth was held by the MIDDLE of the distribution." The two most common categories of thoughts listed were "total wealth held by the richest group" (210 instances) and "difference between the richest and poorest groups" (178 instances). The least frequently mentioned category was "middle of the distribution" (52 instances).

Taken together, Studies 6–8 suggest that people pay less attention to the middle compared to the extremes of an economic distribution, independent of their political preferences and stated values. We suggest that one way to overcome middle neglect is to display economic distributions visually rather than in tabular or text formats. Visual displays have been found in previous research to facilitate more simultaneous and even-handed processing of complex information[40].

In Study 9, we examined whether middle neglect can be attenuated by displaying the same information visually rather than tabularly. We presented 393 participants with a pair of wealth distributions, both divided into four quartiles. One of these distributions had a larger gap between the most- and the least-wealthy quartiles but a smaller gap between intermediate quartiles (a "Smoother" distribution). The other distribution had a smaller gap between the most- and the least-wealthy quartiles but a larger gap between the intermediate quartiles (a "Smaller Top-Bottom Ratio" distribution). Participants were randomly assigned to either a "Tabular" condition in which this information was presented numerically (see Fig. 3A; $N = 200$) or a "Graphical" condition in which this information was presented visually (see Fig. 3B; $N = 193$).

We asked participants to select the distribution that seems more fair to them. Consistent with our prediction, we found that participants were more than twice as likely to choose the Smoother distribution in the Graphical condition (30%) than in the Tabular condition (13%, $\chi^2(1, 393) = 15.14$, $p < 0.001$, $b = 1.03$, 95% CI = [0.52, 1.55]). Thus, people appear to afford relatively more weight to (in)equality among middle groups when information is presented visually than tabularly, and they appear to focus more selectively on top-bottom comparisons when information is presented in tabular formats. Because the Gini Coefficient and other measures of inequality are so sensitive to the wealth of intermediate fractiles, visual displays can thus shift stated inequality preferences markedly by increasing attention to the middle.

## Discussion

In this article we have shown how the most common depictions of economic distributions have features that can bias people's impressions about the extent of economic inequality. We have attributed these biases to the simplifying strategies that people use when evaluating such complex information.

Studies 1–5 provide evidence that when people evaluate an economic distribution, they exhibit *partition dependence* such that their judgments are influenced by how the population happens to be divided into groups. People act as if they compare a given distribution against a benchmark of equality between the identified population groups, with insufficient sensitivity to differences in group sizes. Whenever media outlets display economic distributions in ways that partition the top of the distribution into more (or finer) sub-groups compared to the bottom of the distribution, this decision highlights the extreme amount held by the wealthiest fractions of a population but inadvertently leads readers to underestimate the overall extent of inequality. Indeed, this bias may be exacerbated when resources are especially concentrated at the top of the distribution, as is often the case[41].

Studies 6-9 demonstrate that when people evaluate economic distributions, they also exhibit *middle neglect*—a tendency to overweight information about the wealthiest and least wealthy groups (compared to middle groups) due to their relative salience. We provide evidence that middle neglect is a consequence of how people

A(i): Smoother Distribution

| Percent of Population | Percent of Wealth They Own |
|---|---|
| Richest 25% of people | 50% |
| Next 25% of people | 29% |
| Next 25% of people | 18% |
| Poorest 25% of people | 3% |

A(ii): Smaller Top−Bottom Ratio

| Percent of Population | Percent of Wealth They Own |
|---|---|
| Richest 25% of people | 40% |
| Next 25% of people | 38% |
| Next 25% of people | 12% |
| Poorest 25% of people | 10% |

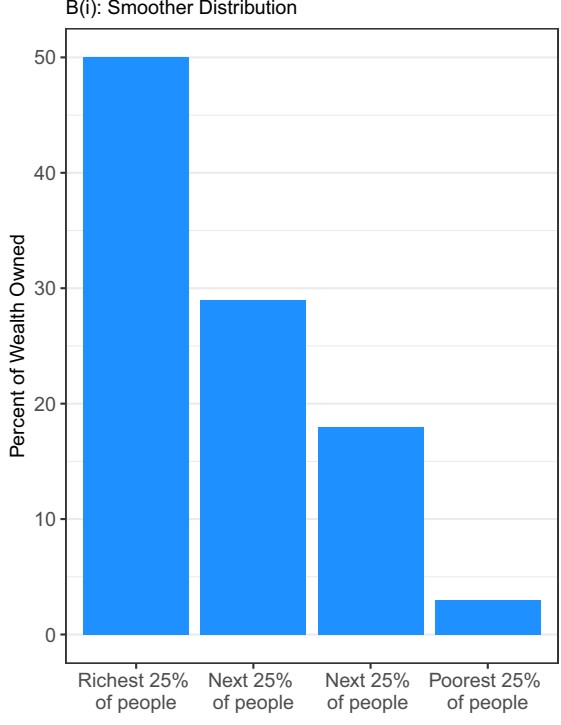

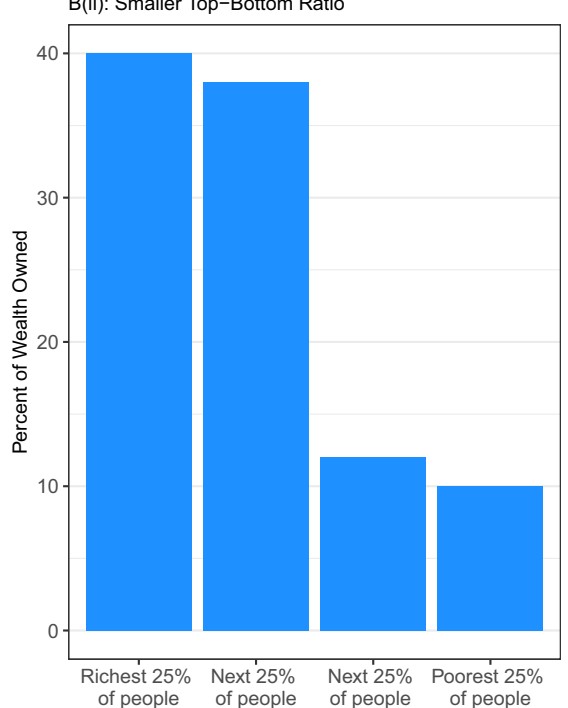

**Fig. 3 | Study 9 Stimuli: Smooth *versus* Polarized Distributions, Graphs versus Tables.** In Study 9, participants chose which distribution seemed more fair when the options were presented either as a table (**A**) or a graph (**B**). Note that the first distribution is smoother whereas the second distribution has a smaller ratio of top-to-bottom wealth. In tabular form, participants seemed to give more weight to the top and bottom quartiles (and less to the intermediate quartiles) compared to when the same information was displayed graphically.

cognitively simplify information when evaluating economic distributions and does not reflect a principled belief that the amount of wealth held by the middle groups is unimportant. As a result, middle neglect can also lead people to form judgments about economic distributions that are shaped by the ways in which journalists happen to divide the population and, thereby, define the "middle" groups.

We acknowledge that all studies in the present investigation were conducted using only American participants. While we presume that the biases we document are common to people across varying circumstances, future work should address this limitation. In particular, future research might evaluate the extent to which these biases persist among people from very different cultures. People may develop differing attitudes about inequality as a result of their experience with different economic structures or cultures (e.g., more collectivist societies). For instance, the myopic focus on just the top and bottom of an economic distribution may vary by culture. Another limitation of the present investigation is that it only considers how explicit information shapes people's impressions of inequality; as noted, this is only one source of people's beliefs about economic inequality. Future research might consider how much, globally, people's views are influenced by statistics they encounter (as in the present studies) versus other information sources such as firsthand experiences.

### Minimizing biases in evaluations of economic inequality

It is important to consider people's susceptibility to partition dependence and middle neglect when choosing how to disseminate information about economic inequality. As documented in our audit of 2958 newspaper articles, popular accounts of economic inequality typically promulgate information in formats that can inadvertently mislead readers and may thereby undermine well-informed public discourse.

In this research, we have identified methods of presenting economic distributions that can attenuate both partition dependence and middle neglect. First, partition dependence can be mitigated by presenting information about economic distributions using partition-invariant metrics such as average wealth per person in each population group. For instance, Study 5 found that partition dependence was reduced substantially by using scaled metrics. Second, middle neglect can be mitigated using visual displays that allow viewers to apprehend at a glance more information across all groups. For instance, Study 8 showed that visual displays of economic distributions can increase sensitivity to differences in the economic standing of intermediate groups.

Both remedies can be simultaneously implemented by using *proportional visual displays* ("PVDs"), as illustrated in Fig. 4B (compared to Fig. 4A). For a related approach, see Page and Goldstein[42]. PVDs use a partition-invariant metric to depict the amount of wealth

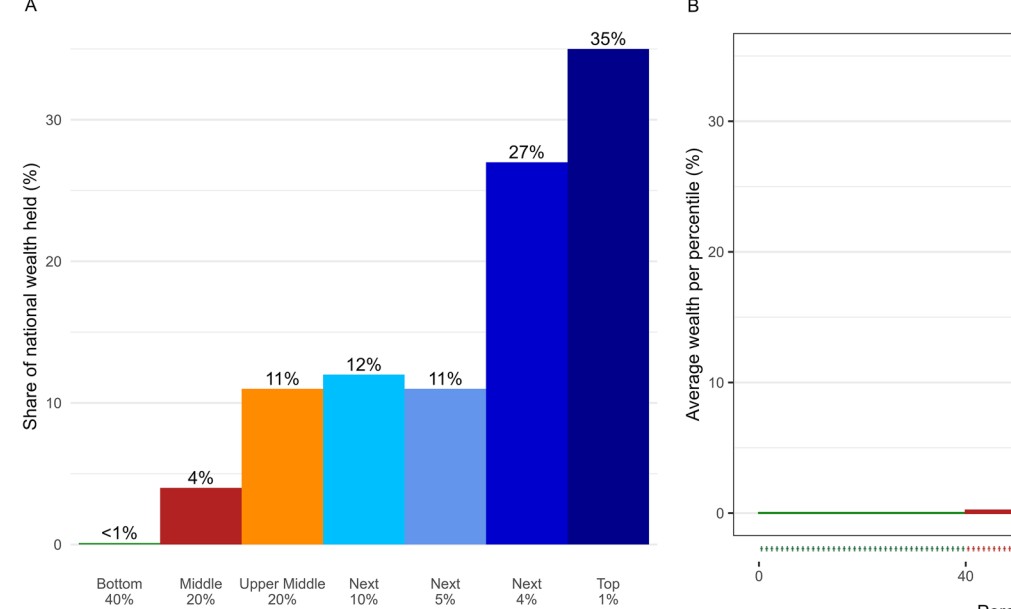

**Fig. 4 | Common *versus* Improved Displays of American Distribution of Wealth. A** provides an example of how wealth inequality is typically represented in the popular press, using a metric of "share of national wealth held by each (unequally sized) group." This image was adapted from one such popular display. **B** presents the same distribution depicted as a proportional visual display (PVD). We assert that the former display leads people to underestimate the extent of inequality because they fail to rescale the share of national wealth (i.e., the height of the bars) according to the varying sizes of the population groups (i.e., the width of the bars). In contrast, the latter depiction leads to more evenhanded evaluation since it accounts for group size, and wealth is thus presented in a partition-invariant metric (average wealth per percentile).

held by each population group (i.e., the height of the bars in Fig. 4B), and the size of each population group is represented visually (i.e., the width of the bars in Fig. 4B). These features may reduce both partition dependence and middle neglect, leaving readers with more accurate impressions of the overall economic distribution.

## Implications

Choosing a format for displaying an economic distribution may appear to be an inconsequential decision for journalists and editors. Yet we have shown that these seemingly trivial choices can have a substantial impact on people's perceptions of inequality and their policy preferences. In fact, we found in our studies that the presentation format of an economic distribution affected support for redistributional policies about as strongly as political affiliation. (See Supplementary Information Tables S2, S3, S12 for details.) As we show in our audit of popular media portrayals of economic distributions, the most common ways of representing inequality tend to bias people toward *underestimating* true levels of inequality, thereby undercutting support for such policies. We suspect that the countless media articles, documentaries, speeches, and books that present economic distributions in biasing formats may have a profound cumulative impact on public impressions of inequality, which in turn may affect policy debates and even election outcomes. For a helpful framework describing how subjective inequality perceptions can affect policy attitudes, see discussion by Jachimowicz and colleagues[43]. Indeed, others have shown that cumulative exposure to media portrayals of inequality impacts people's perceptions of economic fairness[44].

Our results may also have implications beyond the domain of economic inequality. The psychological insights we have investigated in this article may apply broadly to people's judgments about various kinds of finite resource distributions. For instance, we would expect to observe partition dependence and middle neglect when people encounter Alan Krueger's observation that, "The top 5 percent [of music performers] take home 85 percent of all concert revenues."[45] Likewise, these biases may affect people's judgments about the distribution of greenhouse gas emissions by country, National Merit

scholarships by school district, stock options within a company, virus infection rates by state, and so on.

In modern political and professional life, people are often confronted with information about the distribution of finite resources across populations. We are commonly asked to make judgments about how money, opportunities, burdens, or resources ought to be divided across employees, school districts, states, and nations. Such tasks are deceptively demanding, and the cognitive strategies we tend to employ to simplify these tasks can systematically bias our judgment. It is thus critical that communicators recognize and accommodate these constraints in human judgment when disseminating information about inequality. As we have shown, prevailing methods for presenting economic distributions collide with limitations in human reasoning about abstract, large-scale resource distributions. Fortunately, we have also provided a solution. The use of partition-invariant metrics and proportional visual displays can characterize distributional data in ways that promote more accurate understanding and therefore better-informed public discourse.

## Methods

All research conducted for this project complied with all relevant ethical regulations and was approved by the Institutional Review Board at the University of California, Los Angeles. All study pre-registrations were posted before their respective launch date, and materials, data, and pre-registrations can be found via the ResearchBox collection housed in Zenodo. The numbers reported below represent the final sample after all pre-registered exclusions apply (chiefly, attention checks). Sample sizes were determined by back-of-the-envelope estimation of effect size and standard deviations. Unless otherwise noted, all results below are derived from ordinary least squares regressions, with variables defined in each section. See the Supplementary Information (Table S14) for results of regression assumption tests. Unless otherwise noted, variables corresponding to randomly assigned experimental conditions have been dummy coded. In all studies, participants provided informed consent before participating in the survey and were paid at wages consistent with prevailing wages for online

survey participation at the time. We collected participants' self-reported gender identity but, absent theoretical motivation, did not include gender in our analyses.

## Study 1: Scaled metrics and uneven bins
In Study 1 (pre-registration: https://aspredicted.org/hfsg-hpcw.pdf), we recruited 391 participants ($N = 391$; $n_{control} = 195$, $n_{treatment} = 196$) from Amazon's Mechanical Turk platform who met pre-registered inclusion criteria (53% female, $M_{age} = 34.1$, $SD_{age} = 13.1$). As in all studies, surveys were administered via Qualtrics and data were analyzed using R. Participants were graphically shown how wealth is distributed between the top 10% and bottom 90% of Americans. We randomly assigned participants to see this distribution using either an aggregate (i.e., total wealth) or scaled (i.e., average wealth per person) metric, and asked participants about the extent to which they would support a policy that redistributes wealth from richer groups to poorer groups ($-5$ = strongly oppose, $+5$ = strongly support). Consistent with our hypothesis, we observed greater support for redistribution among those who viewed the same distribution expressed in a scaled metric ($t(389) = 2.60$, $p = 0.010$, $b = 0.89$, 95% CI = [0.22, 1.57]). There were not significant effects of the manipulation on participants' judgments of the fairness of the distribution ($t(389) = -1.78$, $p = 0.076$, $b = -0.52$, 95% CI = [$-1.11$, 0.06]), despite the significant effect on their support for redistribution. See the Supplementary Information for further discussion. Considering simply the valence of participants' responses—a hint of how a person might vote on a forced-choice binary ballot measure—we found that just 56% (versus fully 71%) of respondents supported a policy to redistribute wealth when the distribution was represented using an aggregate (versus scaled) metric ($\chi^2(1) = 8.23$, $p = 0.004$, $b = 0.16$, 95% CI = [0.05, 0.26]).

## Study 2: Partition dependence
In this study (pre-registration: https://aspredicted.org/j4w7-zhyn.pdf), we recruited 165 participants from Amazon's Mechanical Turk who met the inclusion criteria (44% female, $M_{age} = 38.0$, $SD_{age} = 9.5$). We pause to highlight that, per our pre-registrations, in all studies of partition dependence—Studies 2 – 5—we begin each session with a rigorous training and practice session (see materials) to familiarize participants with the key concepts required to meaningfully evaluate economic distributions. After all of this, participants were given a brief comprehension quiz which, as pre-registered, they were required to pass in order to be included in analyses. This typically results in a loss of ~25% of the sample who otherwise pass all other exclusion criteria. Critically, this exclusion occurs *prior* to treatment randomization, so no differential attrition is possible. Furthermore, because this sample has been thoroughly trained in the critical concepts, and should therefore be less likely to make mistakes related to partition dependence, we take ours to be an especially conservative set of tests of our hypotheses.

Participants were randomly assigned to construct the actual and ideal wealth distribution of the United States using one of two population partitions. In the Quintiles condition, participants used population quintiles (richest 20% of Americans, next 20%,..., poorest 20%), and in the Quasi-logarithmic condition the population groups were defined as richest 1%, next 4%, next 15%, next 30%, poorest 50% (Fig. 2A, B). In each condition, participants indicated what they believed to be the *current* wealth distribution in the United States (Descriptive Allocation Task) as well as their opinion on how wealth ideally should be distributed (Ideal Allocation Task). As our key outcome variable, we measured the share of total national wealth that participant allocated to the top quintile in each condition.

## Study 3: Different number of groups
In this study (pre-registration: https://aspredicted.org/z2fm-kntc.pdf), we tested the generalizability of partition dependence in evaluations

of economic distributions using alternative methods of presenting this information. We wanted to ensure that the effects of partitioning were not unique to an uneven, quasi-logarithmic unpacking of the top wealth group. Therefore, in this study we unpacked the top wealth group into evenly spaced population groups.

In Study 3, we recruited 162 participants from Amazon's Mechanical Turk who met the inclusion criteria (51% female, $M_{age} = 37.7$, $SD_{age} = 11.8$). We note that, due to a failure in the recruitment process, we only obtained $n = 121$ and $n = 124$ for the two experimental conditions, rather than the pre-registered $n = 125$/condition. Following the same paradigm as in Study 1, participants completed the Descriptive and Ideal Allocation Tasks using one of two population partitions. In the {50-50} condition, participants simply estimated how much wealth was held by the "richest 50% of Americans" and the "poorest 50% of Americans," as well as how much wealth they thought each of these groups ideally should hold. In the {10-10-10-10-10-50} condition, the wealthiest 50% of Americans were partitioned into five deciles. Our key outcome variable was the share of total national wealth allocated to the wealthiest 50% of the population.

## Study 4: Actual distribution given
In this study (pre-registration: https://aspredicted.org/69fz-qpv6.pdf), we focused on the Ideal Allocation Task and tested whether partition dependence would be observed when people were provided with accurate information about the current distribution.

In Study 4, we recruited 475 participants from Amazon's Mechanical Turk who met the inclusion criteria (44% female, $M_{age} = 39.0$, $SD_{age} = 12.9$). Participants were presented with accurate information on the current wealth distribution in the United States (using a 50-50 population partition). Then, participants were randomly assigned to either a Deciles condition or Quartiles condition to indicate their ideal wealth distribution (using the same Ideal Allocation Task as in Studies 1 and 2). In the Deciles condition, the population was partitioned into six groups (richest 10%, next 10% next 10%, next 10%, next 10%, poorest 50%). In the Quartiles condition the population was partitioned into three groups (richest 25%, next 25%, poorest 50%). Our key outcome variable was the share of total national wealth allocated to the wealthiest 50% of the population.

## Study 5: The effect of partition-invariant metrics
This study tested a method to diminish partition dependence. In Study 5 (pre-registration: https://aspredicted.org/zfsv-zqsr.pdf), we recruited 679 participants from Amazon's Mechanical Turk who met the inclusion criteria (56% female, $M_{age} = 37.7$, $SD_{age} = 12.0$). Participants were randomly assigned to one of four experimental conditions in a 2 (partitioning: quintiles versus quasi-logarithmic) × 2 (units: total versus average wealth) experimental design. The first experimental factor varied whether participants were presented with population groups partitioned either evenly into quintiles (richest 20%, next 20%, ..., poorest 20%) or quasi-logarithmically (richest 1%, next 4%, next 5%, next 10%, next 20%, ..., poorest 20%). The second experimental factor varied the metric used to define the amount of wealth held by each population group. Half of participants were presented with the distribution using a Total Wealth metric (i.e., displaying the percentage of total wealth held by a given population group), and the other half of participants were presented with an Average Wealth metric (i.e., the average wealth held per percentile; or, as we used, the percent of total wealth held per percentile). Average Wealth metrics are partition invariant since they are scaled by the number of individuals in a group.

Participants observed the wealth distribution and then responded to three questions asking their opinions on the fairness of the distribution and three questions asking about the extent to which they would support redistributing wealth from citizens who are richer to those who are poorer. The order of the three questions within each

block was randomized. Participants were shown a given distribution and asked to place themselves on the following scales:

Fairness item 1 (11-point scale): 1 = Ideally, rich Americans would own much more wealth, 6 = Ideally, no change from this distribution, 11 = Ideally, poor Americans would own much more wealth.

Fairness item 2 (11-point scale): 1 = Ideally, the wealth distribution should be much more unequal, 6 = Ideally, no change from this distribution, 11 = Ideally, the wealth distribution should be much more equal.

Fairness item 3 (7-point scale): 1 = This distribution is not at all unfair, 7 = This distribution is extremely unfair.

Policy support item 1 (11-point scale): 1 = Given this distribution, I would strongly oppose a government policy that redistributes more money to the poor, 6 = Neither oppose nor support, 11 = Given this distribution, I would strongly support a government policy that redistributes more money to the poor.

Policy support item 2 (11-point scale): 1 = Given this distribution, I think the government should significantly decrease taxes on the rich, 6 = Taxes on the rich should stay the same as they are today, 11 = Given this distribution, I think the government should significantly increase taxes on the rich.

Policy support item 3 (11-point scale): 1 = Given this distribution, I think the government should significantly decrease services that benefit the poor, 6 = Government services that benefit the poor should stay the same as they are today, 11 = Given this distribution, I think the government should significantly increase services that benefit the poor.

From these six questions we constructed composite measures of fairness ($\alpha = 0.70$) and support for redistribution ($\alpha = 0.88$). We pre-registered a prediction that the partition dependence bias would we be diminished (i.e., a smaller difference between the population partitions in terms of reported fairness and support for redistributions) when wealth is expressed using a partition-invariant metric (i.e., average wealth).

## Study 6: Middle neglect

We recruited participants from Amazon's Mechanical Turk and, after removing participants who failed pre-registered inclusion criteria and three others who completed the entire survey in under 10 seconds, we were left with 197 participants (41% female, $M_{age} = 34.6$, $SD_{age} = 11.4$; pre-registration: https://aspredicted.org/93zd-x2b6.pdf). Participants were asked to make a series of judgments about the economic distribution of eight unnamed Western, democratic countries. Each economic distribution displayed the average annual income of three population segments: the richest quintile of society, the middle quintile, and the poorest quintile. We varied the average income for each population segment to be either relatively low or high. Our goal was to measure how independent changes to the income held by each population segment influenced judgments of fairness and support for redistribution. The low income-level for each population segment is roughly equal to the actual average annual income for the richest, middle, and poorest quintiles of the United States population as of 2020. The high income-level was defined by simply doubling that of the low level. Specifically, we varied the economic distributions such that the richest quintile of society was described as having an average annual income of $220,000 or $440,000; the middle quintile was described as having an average income of $60,000 or $120,000; and the poorest quintile was described as earning and average of $14,000 or $28,000 (Table S4). Participants observed all combinations of income levels in a random order, which yielded a total of eight hypothetical economic distributions (i.e., two income levels for three population segments in a $2^3$ factorial within-participants design). Participants were asked to assume that each distribution represented a Western, democratic country that was roughly equal in terms of population size, demographics, and total national wealth. While

observing each distribution sequentially, participants rated the extent to which they thought each distribution was fair (−5 = completely unfair, +5 = completely fair) as well as their support for a government policy to redistribute wealth (−5 = strongly oppose, +5 = strongly support). After rating all eight combinations of average incomes, participants responded to a series of items asking their opinions on economic inequality. Participants rated the extent to which they agreed or disagreed with the following statements (−7 = completely disagree, 0 = neither agree nor disagree, 7 = completely agree): "We ought to make sure that the POOREST members of society are doing as well as they can"; "We ought to build as strong, robust and wealthy of a MIDDLE class as possible"; and "We ought to make sure that it is possible to become EXTREMELY WEALTHY to incentivize people." Lastly, participants completed the same set of questions on political attitudes as in previous studies.

## Study 7: Testing the mechanism of middle neglect

We surveyed 868 participants from Amazon's Mechanical Turk who met pre-registered inclusion criteria (45% female, $M_{age} = 41.7$, $SD_{age} = 13.1$; pre-registration: https://aspredicted.org/x2n5-qhsh.pdf). Participants were shown sixteen hypothetical economic distributions—corresponding to all $2^4$ combinations of a low and high value of wealth held by each of four wealth quartiles—displayed in a tabular format. We randomly assigned participants to one of three between-participants conditions in which we (a) visually highlighted the richest and poorest groups (i.e., the top and bottom rows in the table were highlighted in yellow), (b) visually highlighted the middle groups (i.e., the upper-middle and lower-middle groups), or (c) highlighted none of the rows. For each distribution shown, we asked participants to rate the fairness of that distribution.

We pre-registered our hypothesis that highlighting the *extremes* of an economic distribution would make a smaller difference in sensitivity to changes of the values in the *extremes* (versus no highlighting) compared to the relatively larger effect of highlighting the *middle* groups on sensitivity to changes in the *middle* (versus no highlighting). To test this, we estimated an OLS regression predicting fairness judgment from the fully saturated model corresponding to the interaction of: (1) the within-participant experimental factor of low-versus-high levels for all four wealth quartiles, and (2) two indicator variables corresponding to the between-participants experimental condition of seeing the middle or extreme quartiles highlighted (thus, the control condition served as the reference category). Additionally, we controlled for stated concern for the economic well-being of each economic quartile. Standard errors were clustered at the individual-level. Analytically, our hypothesis comes to the prediction that: (a) the average of the regression coefficients of [poor_high_wealth*highlight_extremes and rich_high_wealth*highlight_extremes] would be smaller than (b) the average of the regressions coefficients of [lower-middle_high_wealth*highlight_middle and upper-middle_high_wealth*highlight_middle]. We estimated the linear combinations of these coefficients then bootstrapped 10,000 samples to estimate the mean difference of the foregoing comparison in order to construct a 95% confidence interval. As predicted, highlighting the extreme groups had a small effect on fairness judgements, whereas highlighting the middle groups had a much larger effect ($b = 0.083$, Bootstrapped 95% CI = [0.05, 0.11]). We also tested a related comparison: (a) [how much more highlighting extremes increases sensitivity to the extremes (compared to no highlighting) than it increases sensitivity to the middle (compared to no highlighting)] *versus* (b) [how much more highlighting the middle increases sensitivity to the middle (compared to no highlighting) than it increases sensitivity to the extremes (compared to no highlighting)]. We find a similar estimated effect ($b = 0.093$, Bootstrapped 95% CI = [0.05, 0.13]). These results are consistent with a "cognitive blind spot", rather than ideological, explanation for the phenomenon of middle neglect.

## Study 8: Thought-elicitation procedure to examine middle neglect

We recruited 269 participants from Amazon's Mechanical Turk who met pre-registered inclusion criteria (53% female, $M_{age}$ = 38.3, $SD_{age}$ = 12.0). Participants were asked to list the thoughts that occurred to them while observing the economic distributions for an unnamed country. To ensure that our results were not being driven by the particularities of a given distribution or the way it was presented, participants were randomly assigned to observe one of six possible distributions corresponding to a 2 (distribution: highly unequal versus highly equal) × 3 (presentation: table, pie chart, bar chart) between-participants design. Participants evaluated the economic distribution on fairness and considered their support for redistribution, as in previous studies. Next, participants listed each of the thoughts that occurred to them as they made these evaluations. Participants could list up to seven thoughts as open-ended text (number of thoughts listed: $M_{thoughts}$ = 3.2, $SD_{thoughts}$ = 1.4). Lastly, participants self-coded each of their listed thoughts into one of the following seven categories:

1. I focused on how much of the total wealth was concentrated at the TOP of the distribution
2. I focused on how much of the total wealth was at the BOTTOM of the distribution
3. I focused on the DIFFERENCE between the amount of wealth at the TOP vs the BOTTOM of the distribution
4. I focused on the RATIO of wealth at TOP to wealth at the BOTTOM of the distribution
5. I focused on how much of the total wealth was held by the MIDDLE of the distribution
6. I focused on how much the distribution differed from total equality across groups
7. Other

If participants selected "other," they were asked to create their own label. Using this thought-elicitation data, we observed the extent to which people reported focusing on each part of the distribution.

## Study 9: mitigating middle neglect with visual displays

In Study 9 (pre-registration: https://aspredicted.org/6444-hxkk.pdf), we recruited 393 participants from Amazon's Mechanical Turk who met pre-registered inclusion criteria (50% female, $M_{age}$ = 39.2, $SD_{age}$ = 11.6). Participants observed a pair of wealth distributions for unnamed countries with the populations partitioned into even quartiles. One of these distributions had a larger gap between the most- and the least-wealthy quartiles but a smaller gap between intermediate quartiles ("Smoother"). The other distribution had a smaller gap between the most- and least-wealthy quartiles but a larger gap between the intermediate quartiles ("Smaller Top-Bottom Ratio"). Participants were randomly assigned to observe the distributions presented either as tables or as bar graphs. While observing the pair of distributions, participants made a binary choice: "Which of these two distributions seems to you to be more fair?"

We pre-registered a prediction that participants would be more likely to choose the Smoother distribution when these distributions were displayed as graphs (versus tables) since a visual display more easily facilitates evaluation of the full distribution.

### Reporting summary

Further information on research design is available in the Nature Portfolio Reporting Summary linked to this article.

## Data availability

All data for the experiments have been anonymized and are publicly available in a ResearchBox collection stored in a Zenodo repository (https://doi.org/10.5281/zenodo.15492434).

## Code availability

Analysis code for all experiments are also publicly available in the ResearchBox collection in the Zenodo repository (https://doi.org/10.5281/zenodo.15492434).

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

## Acknowledgements

The authors would like to thank Madison Thantu for expertly guiding the team of RAs in conducting the newspaper audit, and also Dave Nussbaum for comments on an early draft of this manuscript that were equal parts thoughtful and helpful.

## Author contributions

J.B.: Conceptualization, Data curation, Formal analysis, Investigation, Methodology, Visualization, Writing—original draft, Writing—review & editing; C.W.: Conceptualization, Data curation, Formal analysis, Investigation, Methodology, Visualization, Writing—original draft, Writing—review & editing; C.F.: Conceptualization, Funding acquisition, Investigation, Methodology, Writing—original draft, Writing—review & editing.

## Competing interests

The authors declare no competing interests.
