## [Transparent Peer Review file · Nature Communications]

How Common Depictions of Wealth Distributions Can Bias People to Underestimate Inequality

Corresponding Author: Dr Jonathan Bogard

Version 0:

Reviewer comments:

Reviewer #1

(Remarks to the Author)

I reviewed a previous version of this paper; I believe the authors did a great job in reviewing their manuscript, and I am satisfied with their replies to my comments. I think their work is really interesting and worth publishing.

Reviewer #2

(Remarks to the Author)

As someone who reviewed an earlier version of this paper, I very much enjoyed reading the authors' responses and re-reading the entire paper. I applaud the authors for thoroughly revising their manuscript, and have just a few more (relatively minor) remarks:

- When describing the analyses of media outlets, it might be helpful to give one or two concrete examples of articles (when and where they were published, how they depicted inequality, etc.)
- In Study 3, participants were asked to report both their ideal and their estimated levels of inequality, based on different partitions. Is the gap between participants' ideal and estimated larger under one partition or another? If so, this may reflect another problem that is propagated by typical media depictions
- The example of "middle neglect" given in page 11 ("The 3 richest Americans...") is not a great one, since it refers to a partition that actually includes the middle ("...bottom 50%). Do they have better examples for this?
- On page 12, the authors report the results from a scale that goes from -7 to +7. However, the results are (I believe) reported on a 1 to 15 scale. Might be better to report these in the same way they were collected (e.g., a mean of 8 should be 0).
- I really appreciated the addition of Study 7. Could the authors please include the average fairness in each condition?
- Relatedly, I wondered why Study 7 confounded emphasizing the salience of the top and the bottom in the same condition. Wouldn't it be much more informative to separate this into two different conditions? In addition to testing the effect of highlighting one group, this would also solve a confound in the study (confounds what is being highlighted with how many rows are being highlighted).
- One last thing: The title and abstract of the papers still heavily rely on the notion of bias. I am worried that by using this term, readers might get the impression that there is a 'right' and a 'wrong' way of thinking about inequality. Is that true? Wouldn't a more accurate description be "Common depictions of economic inequality evoke underestimations of its scope?" or something like that?

-

Reviewer #4

(Remarks to the Author)

I read this paper with great interest. The authors show convincingly that (1) current portrayals of inequality in the media may induce several biases and distortions, (2) show several systematic patterns with which those biases appear across various depictions of inequality and (3) provide some suggestions on how to improve the presentation to avoid biases/distortions.

While I was not part of the original reviewer team and therefore can only look at this version of the paper and not the initial submission, I find this paper clear, well written, and offering several novel results. This is a great paper!

Contribution: This work builds nicely on the (now wide-ranging) literature on misperceptions of inequality spearhead by Norton & Ariely (2011). By going beyond documenting misperceptions, this paper offers some systematic and novel insights by showing which parts of the distribution people pay particular attention to and how media, and even academics, might want to do a better job presenting inequality. This nicely complements recent work by Blesch et al. (NHB, 2022) who argue that researchers and policy-makers should pay more attention to where inequality is concentrated (beyond the Gini). This paper here makes a nice contribution to this field with clear and practical guidance.

Methods and results: I found the combination of the media audit study alongside the pre-registered experiments very convincing, both ensuring external validity of this research while also focusing on causality and mechanism. The statistical results are clear and robust across a variety of studies and methods. Given that the authors have already addressed several concerns in the response to the reviewers, I didn't find any major problems here; I think that this is a strong and robust set of results.

Literature: The only area I found a bit lacking is the related literature: there are now many papers on these questions, and it seems the authors have focused their literature review more on "seminal" papers, and less on recent ones. Some examples that I'd recommend to include are the following, but I'd recommend looking more widely for other related and ongoing work that's come out in the last 1-2 years, so as to position this paper (and its contribution) within the wider context of this active field of research:

- Davidai & Wienk (2021). The psychology of lay beliefs about economic mobility. SPPC.
- Diermeier et al. (2017). Impact of inequality-related media coverage on the concerns of the citizens. Working Paper.
- Jachimowicz et al. (2022). Inequality in researchers' minds: Four guiding questions for studying subjective perceptions of economic inequality. J Econ Surveys.
- Phillips et al. (2020). Inequality in people's minds. Working paper.

Version 1:

Reviewer comments:

Reviewer #4

(Remarks to the Author)

I'm happy for the paper to be published in its current form.

(Remarks on code availability)

Reviewer #1:

I reviewed a previous version of this paper; I believe the authors did a great job in reviewing their manuscript, and I am satisfied with their replies to my comments. I think their work is really interesting and worth publishing.

- **Thank you for your help in improving our work in previous revisions!**

Reviewer #2:

As someone who reviewed an earlier version of this paper, I very much enjoyed reading the authors' responses and re-reading the entire paper. I applaud the authors for thoroughly revising their manuscript, and have just a few more (relatively minor) remarks:

- When describing the analyses of media outlets, it might be helpful to give one or two concrete examples of articles (when and where they were published, how they depicted inequality, etc.)

- **We have included an example. Thanks for this suggestion.**

- In Study 3, participants were asked to report both their ideal and their estimated levels of inequality, based on different partitions. Is the gap between participants' ideal and estimated larger under one partition or another? If so, this may reflect another problem that is propagated by typical media depictions

- **This is a really interesting question to ask, and something we would not have predicted based on our conceptualization. While we are reluctant to include *post hoc* analyses that (a) were not pre-registered and (b) are not central to our hypotheses or derived from our analyses, we ran the analysis out of curiosity! It turns out that the difference between participants' (stated) preferred distribution and their estimate of the actual distribution does *not* significantly differ by condition.**

- The example of "middle neglect" given in page 11 ("The 3 richest Americans...") is not a great one, since it refers to a partition that actually includes the middle ("...bottom 50%). Do they have better examples for this?

- **Thanks for pointing this out! We have both changed the example and added both an explication of the example as well as a general clarification that "middle neglect" is about neglecting intermediate population groups (not just neglecting the 50th percentile). Unfortunately, given how vast inequality is both within the United States and between countries, it is impractical to compare, for instance, the top 10% and the bottom 10% of the US wealth distribution. The former group holds most of the wealth in the US whereas the latter actually holds negative wealth (i.e., debt) on average.**

- On page 12, the authors report the results from a scale that goes from -7 to +7. However, the results are (I believe) reported on a 1 to 15 scale. Might be better to report these in the same way they were collected (e.g., a mean of 8 should be 0).

- **Changed! Thanks for catching that.**

- I really appreciated the addition of Study 7. Could the authors please include the average fairness in each condition?

- **This is a good suggestion. We have added labels to the plot in the graph depicting the results of Study 7. Because there are so many results to report – a low- and high-income level for each of four population quartiles for each of three conditions (i.e., 24 cell means) – we opted to not report cell means in the manuscript prose but instead just label the graph.**

- Relatedly, I wondered why Study 7 confounded emphasizing the salience of the top and the bottom in the same condition. Wouldn't it be much more informative to separate this into two different conditions? In addition to testing the effect of highlighting one group, this would also solve a confound in the study (confounds what is being highlighted with how many rows are being highlighted).

- **First, we note that there were two experimental conditions in which we highlight rows from the table to add visual salience to certain groups. In in *both* of these conditions, we highlight **two** rows (we highlight the rich and the poor, or we highlight the upper-middle and the lower-middle). Thus between the two experimental conditions, this is held constant. We have clarified this in the manuscript now to avoid misunderstanding. Thanks for pointing out that ambiguity! Second, while we find the question of which group naturally attracts the most visual attention (e.g., the richest or the poorest) an interesting one to consider, it is both beyond the scope of this paper and we have no theoretical basis for predicting one versus the other. Instead, Study 7 is designed simply to offer evidence consistent with our claim that the over-weighting of the most salient information (i.e., the richest and poorest groups) results from a cognitive bias rather than a principled one.**

- One last thing: The title and abstract of the papers still heavily rely on the notion of bias. I am worried that by using this term, readers might get the impression that there is a 'right' and a 'wrong' way of thinking about inequality. Is that true? Wouldn't a more accurate description be "Common depictions of economic inequality evoke underestimations of its scope?" or something like that?

- **We appreciate this suggestion! By "bias" we wish simply to suggest systematic and predictable error. Reviewer 2 is certainly correct that we do not mean to imply that it is biased relative to some objective normative standard. The problem with using a title like the one suggested is that the perceptual biases we identify could just as easily bias people toward *overestimation* of inequality with arbitrary changes to the design of how inequality is depicted. As but one example, unpacking the bottom half of Americans into many groups might do this. In this way, our fundamental hypotheses are more general than specifically claiming underestimation. Nonetheless, Reviewer 2's point is a good one that the most common forms of depiction in the media – as demonstrated by our audit – do bias people toward *underestimating* inequality. If, given all this, the review team still suggests amending the title, we would consider alternatives. Regardless, as a result of this comment, we have added a line to emphasize this point in our "Implications" section of the general discussion to the manuscript. Specifically, we say:**
 - **As we show in our audit of popular media portrayals of economic wellbeing, the most common ways of representing inequality tend to bias people toward *underestimating* true levels of inequality, thereby undercutting support for such policies.**

Reviewer #4:

I read this paper with great interest. The authors show convincingly that (1) current portrayals of inequality in the media may induce several biases and distortions, (2) show several systematic patterns with which those biases appear across various depictions of inequality and (3) provide some suggestions on how to improve the presentation to avoid biases/distortions. While I was not part of the original reviewer team and therefore can only look at this version of the paper and not the initial submission, I find this paper clear, well written, and offering several novel results. This is a great paper!

Contribution: This work builds nicely on the (now wide-ranging) literature on misperceptions of inequality spearhead by Norton & Ariely (2011). By going beyond documenting misperceptions, this paper offers some systematic and novel insights by showing which parts of the distribution people pay particular attention to and how media, and even academics, might want to do a better job presenting inequality. This nicely complements recent work by Blesch et al. (NHB, 2022) who argue that researchers and policy-makers should pay more attention to where inequality is concentrated (beyond the Gini). This paper here makes a nice contribution to this field with clear and practical guidance.

Methods and results: I found the combination of the media audit study alongside the pre-registered experiments very convincing, both ensuring external validity of this research while also focusing on causality and mechanism. The statistical results are clear and robust across a variety of studies and methods. Given that the authors have already addressed several concerns in the response to the reviewers, I didn't find any major problems here; I think that this is a strong and robust set of results.

- **Thanks for all of these comments!**

Literature: The only area I found a bit lacking is the related literature: there are now many papers on these questions, and it seems the authors have focused their literature review more on "seminal" papers, and less on recent ones. Some examples that I'd recommend to include are the following, but I'd recommend looking more widely for other related and ongoing work that's come out in the last 1-2 years, so as to position this paper (and its contribution) within the wider context of this active field of research:

- Davidai & Wienk (2021). The psychology of lay beliefs about economic mobility. SPPC.
- Diermeier et al. (2017). Impact of inequality-related media coverage on the concerns of the citizens. Working Paper.
- Jachimowicz et al. (2022). Inequality in researchers' minds: Four guiding questions for studying subjective perceptions of economic inequality. J Econ Surveys.
- Phillips et al. (2020). Inequality in people's minds. Working paper.

- **Thank you for the suggestion of additional citations. Given the constraints of *Nature: Communications*, we were considerably limited in the past research that we could cite. We have restored several citations that were in previous versions of this manuscript as well as all others you suggested that weren't already included. We're grateful for your pointing out these papers, especially the Diermeier et al. (2017) one – we hadn't seen that working paper and it is obviously relevant.**

We are grateful for the team's thoughtful treatment of this manuscript. In all, we believe we are presenting you with a far higher-quality paper than our original submission, due in large part to the editorial team's extremely helpful, thoughtful comments.

Our gratitude,

The Authorship Team